# Global biogeographic sampling of bacterial secondary metabolism

Zachary Charlop-Powers[1], Jeremy G Owen[1], Boojala Vijay B Reddy[1], Melinda A Ternei[1], Denise O Guimarães[2], Ulysses A de Frias[3], Monica T Pupo[3], Prudy Seepe[4], Zhiyang Feng[5], Sean F Brady[1]*

[1]Laboratory of Genetically Encoded Small Molecules, Howard Hughes Medical Institute, Rockefeller University, New York, United States; [2]Laboratório de Produtos Naturais, Curso de Farmácia, Universidade Federal do Rio de Janeiro–Campus Macaé, Rio de Janeiro, Brazil; [3]School of Pharmaceutical Sciences of Ribeirão Preto, University of São Paulo, São Paulo, Brazil; [4]KwaZulu-Natal Research Institute for Tuberculosis and HIV, Nelson R Mandela School of Medicine, Durban, South Africa; [5]College of Food Science and Technology, Nanjing Agricultural University, Nanjing, China

**Abstract** Recent bacterial (meta)genome sequencing efforts suggest the existence of an enormous untapped reservoir of natural-product-encoding biosynthetic gene clusters in the environment. Here we use the pyro-sequencing of PCR amplicons derived from both nonribosomal peptide adenylation domains and polyketide ketosynthase domains to compare biosynthetic diversity in soil microbiomes from around the globe. We see large differences in domain populations from all except the most proximal and biome-similar samples, suggesting that most microbiomes will encode largely distinct collections of bacterial secondary metabolites. Our data indicate a correlation between two factors, geographic distance and biome-type, and the biosynthetic diversity found in soil environments. By assigning reads to known gene clusters we identify hotspots of biomedically relevant biosynthetic diversity. These observations not only provide new insights into the natural world, they also provide a road map for guiding future natural products discovery efforts.

*For correspondence: sbrady@ rockefeller.edu

**Competing interests:** The authors declare that no competing interests exist.

## Introduction

Soil-dwelling bacteria produce many of the most important members of our pharmacy, including the majority of our antibiotics as well as many of the cytotoxic compounds used in the treatment of cancers (*Cragg and Newman, 2013*). The traditional approach for characterizing the biosynthetic potential of environmental bacteria has been to examine metabolites produced by bacteria grown in monoculture in the lab. However, it is now clear that this simple approach has provided access to only a small fraction of the global microbiome's biosynthetic potential (*Rappe and Giovannoni, 2003*; *Gilbert and Dupont, 2011*; *Rajendhran and Gunasekaran, 2011*). In most environments uncultured bacteria outnumber their cultured counterparts by more than two orders of magnitude, and among the small fraction of bacteria that has been cultured (*Torsvik et al., 1990*, *1998*), only a small subset of gene clusters found in these organisms is generally expressed in common fermentation broths (*Bentley et al., 2002*; *Ikeda et al., 2003*). The direct extraction and subsequent sequencing of DNA from environmental samples using metagenomic methods provides a means of seeing this 'biosynthetic dark matter' for the first time. Unfortunately, the genomic complexity of most metagenomes limits the use of the shotgun-sequencing and assembly approaches (*Iverson et al., 2012*; *Howe et al., 2014*) that are now routinely used to study individual microbial genomes (*Donadio et al., 2007*; *Cimermancic et al., 2014*). Although bacterial natural products represent an amazing diversity of chemical structures, the majority of bacterial secondary metabolites, including most clinically useful microbial metabolites, arise from a

**eLife digest** Many of the most useful medicinal drugs—including antibiotics and cancer drugs—are derived from bacteria living in the soil that produce these chemicals as part of their natural life cycle. Many of these chemicals have been found by culturing bacteria in the laboratory, but this approach is limited because it only provides access to the chemicals produced by the small fraction of bacteria species that we can culture in this way. Also, many bacteria do not produce as many different chemicals when they are grown under these artificial conditions, instead of their natural environment. This suggests that bacteria living in the environment are likely to provide an additional source of new chemicals that could have medicinal benefits.

Here, Charlop-Powers et al. tackle this issue by employing a high-throughput genetic method for assessing the potential of soil-dwelling bacteria to make compounds with biological activity. They extracted DNA directly from soil samples collected from five continents, in part through the efforts of a citizen-science project called 'Drugs from Dirt' (drugsfromdirt.org). These samples came from many different environments, including rainforests, deserts, and coastal sediments.

After extracting the DNA from the soil samples, Charlop-Powers et al. focused on sequencing the genes that encode enzymes called NRPS and PKS. These enzymes are involved in the production of a range of diverse compounds, including many clinically useful antibiotics. By comparing the sequences of the genes found in the different soils, it was possible to estimate how common the genes were in each sample, and also to compare the collections of genes found in different soil types. This comparison revealed that the DNA sequences of the genes encoding NRPS and PKS vary widely among the soil samples, except for samples that came from similar environments in close proximity to each other.

These findings show that populations of soil-dwelling bacteria living in different locations are likely to produce related, but different and largely unexplored, natural compounds that could have the potential to be used in drug therapies or in other industries.

very small number of common biosynthetic themes (e.g., polyketides, ribosomal peptides, non-ribosomal peptides, terpenes, etc) (*Dewick, 2002*). Because of the functional conservation of enzymes used by these common systems, degenerate primers targeting the most common biosynthetic domains provide a means to broadly study gene cluster diversity in the uncultured majority in a way similar to what is now regularly done for bacterial species diversity using 16S rRNA gene sequences. Here we use this approach to conduct the first global examination of non-ribosomal peptide synthetase (NRPS) adenylation domain (AD) and polyketide synthase (PKS) ketosynthase (KS) domain biosynthetic diversity in soil environments. We chose to explore NRPS and PKS biosynthesis because the highly modular nature of these biosynthetic systems has provided a template for the production of a wide variety of gene clusters that give rise to a correspondingly diverse chemical repertoire, including many of the most clinically useful microbial metabolites (*Cragg and Newman, 2013*).

## Results and discussion

With the help of a citizen science effort (www.drugsfromdirt.org), soil samples were collected from five continents (North America, South America, Africa, Asia, Australia) and several oceanic islands (Hawaii, Dominican Republic), covering biomes that include multiple rainforests, temperate forests, deserts and coastal sediments. DNA was extracted directly from these soils as previously described (*Brady, 2007*) and 96 samples were chosen for analysis of NRPS/PKS diversity using 454 pyro-sequencing of AD and KS domain PCR amplicons. Samples were chosen on the basis of DNA quality and biome diversity; raw sequence reads from these samples were combined with existing amplicon datasets derived from other biomes using the same DNA isolation, PCR and sequencing protocols (*Charlop-Powers et al., 2014*). The entire dataset representing 185 biomes was clustered into operational taxonomic units (OTUs) at a sequence distance of 5%. Despite millions of unique sequencing reads yielding a predicted Chao1 OTU estimate of greater than 350,000 for each domain, rarefaction analysis suggests that we have not yet saturated the sequence space of either domain (*Figure 1A,C*).

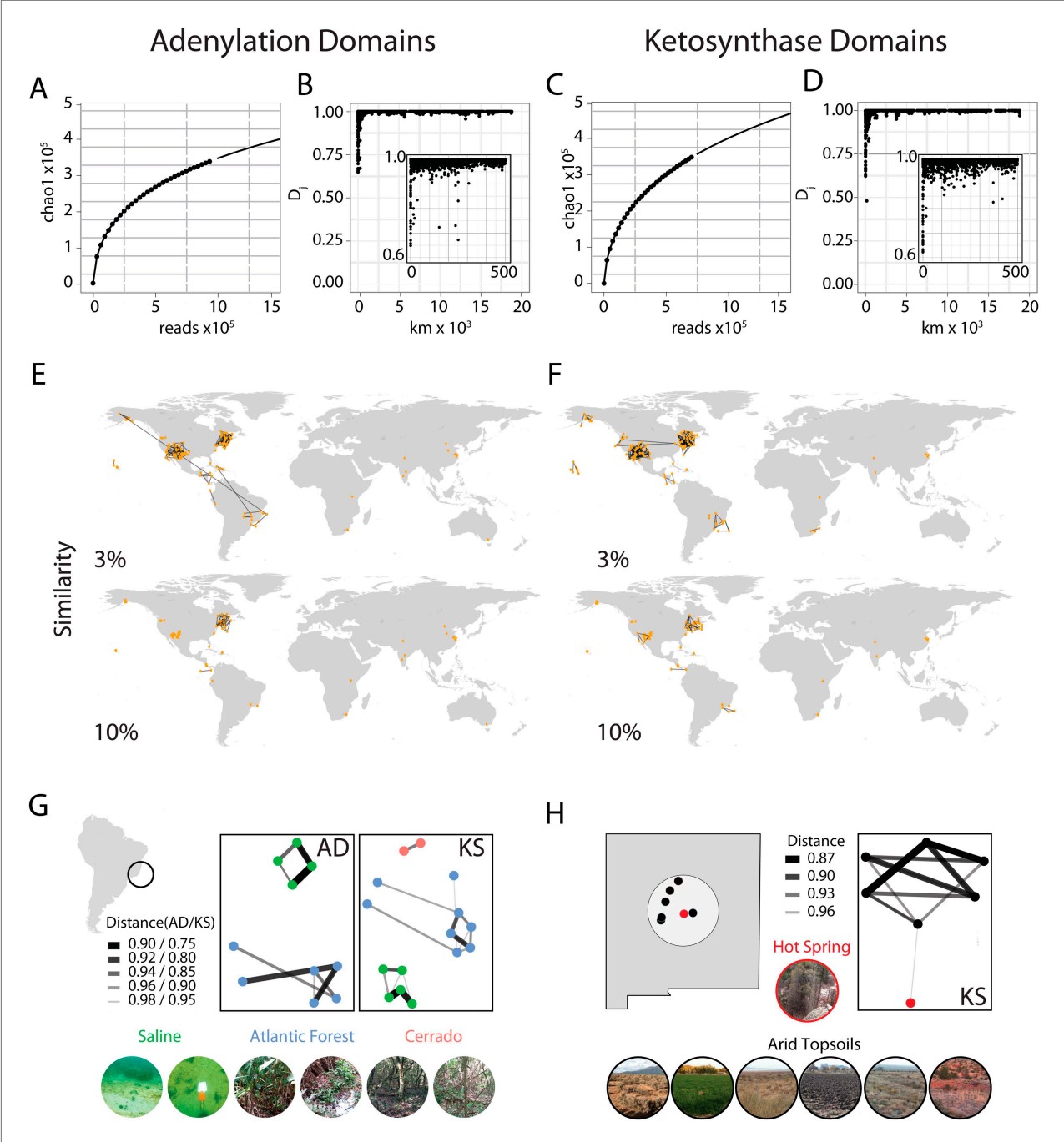

**Figure 1**. Global abundance and comparative distribution of AD/KS sequences. The global abundance (**A** and **C**), sample-to-sample variation (**B** and **D**), and geographic distribution (**E**, **F**, **G**, and **H**) of adenylation domains (AD) and ketosynthase domains (KS) were assessed by pyro-sequencing of amplicons generated using degenerate primers targeting AD and KS domains found in 185 soils/sediments from around the world. (**A** and **C**) Global AD (**A**) or KS (**C**) domain diversity estimates were obtained by rarefying the global OTU table (de novo clustering at 95%) for AD and KS sequences and calculating the average Chao1 diversity metric at each sampling depth. (**B** and **D**) The ecological distance (i.e., Jaccard dissimilarity) between AD (**B**) or KS (**D**) domain populations sequenced from each metagenome was determined as a function of the great circle distance between sample collection sites (km). Insets show local relationships (<500 km) in more detail. (**E** and **F**) All sample collection sites are shown on each world map and lines are used to connect sample sites that share at least the indicated fraction (3%, 10%) of AD (**E**) or KS (**F**) OTUs. (**G** and **H**) Biome-specific relationships within domain OTU populations sequenced from geographically proximal samples assessed by Jaccard similarity. Samples were collected from (**G**) Atlantic forest, saline or cerrado environments or from the (**H**) New Mexican desert topsoils or hot springs sediments.

The first question we sought to address with this data was how biosynthetic sequence composition varies by geographic distance. To do this we calculated the pairwise Jaccard distances between AD/KS sequence sets derived from each sampling site and used these metrics to compare samples. The Jaccard distance, a widely used metric for comparing the fraction of shared OTUs between samples, was chosen over alternative metrics due to its simplicity and to the lack of a comprehensive reference phylogenetic tree for AD and KS domains as exists for 16S analyses. Most Jaccard distances were found to be quite small (<3%), indicating large differences in secondary metabolite gene sequence composition between almost all sample collection sites (*Figure 1B,D*). Although the OTU overlap between our individual experimental samples is generally small, these relationships allow us to begin to develop a picture of how biosynthetic diversity varies globally. On a global level, the strongest biosynthetic sequence composition relationships are seen between samples collected in close physical proximity to one another (*Figure 1B,D,E,F*) as opposed to between samples from similar biomes in different geographic locations. For example, at a cutoff of even as low as 3% shared KS or AD OTUs, essentially all inter-sample relationships are observed between immediate geographic neighbors and not similar biomes in different global locations (*Figure 1E,F*). This likely explains the limited inter-sample relationships we observe between samples from the Eastern hemisphere as most samples from this part of the world were collected from sites at a significant geographic distance from one another. The only exception is the set of soil samples from South Africa, of which a number were collected in relatively close geographic proximity. These samples exhibit similar pairwise Jaccard metrics to those observed between geographically proximal samples collected in the Western hemisphere (*Figure 1E,F*).

Although differences in biosynthetic composition of microbiomes appear to depend at least in part on the geographic distance between samples, our data suggests that change in the biome type is an important additional factor for the differentiation of biosynthetic diversity on a more local level (*Figure 1G,H*). For example, at a cutoff of 3% shared OTUs, essentially all inter-sample relationships are observed between immediate geographic neighbors; when this is raised to 10% shared OTUs (*Figure 1E,F*), relationships are only seen between nearby samples belonging to the same biome. This phenomenon is highlighted by the two examples shown in *Figure 1G,H*. In the first example, Brazilian soils were collected from Atlantic rainforest, saline or cerrado (savanna-like) sites located only a few miles from one another. Our AD and KS data show these sample are (i) distinct from other globally distributed samples, (ii) most strongly related to the samples from the same Brazilian biome and (iii) only distantly related to the samples from other Brazilian biomes. In the second example, a sample collected from a New Mexican hot spring where the soil is heated continuously by subterranean water is compared with samples derived from the dry soils of the surrounding environment. Once again our amplicon data show that these samples are (i) distinct from other globally distributed samples, (ii) most strongly related to other samples from the same biome and (iii) only distantly related to samples from other nearby biomes. Although it is possible that at a much greater sampling depth all AD and KS domains will be found at all sites as predicted by Baas-Becking's 'everything is everywhere but the environment selects' hypothesis of global microbial distribution (*O'Malley, 2007*; *de Wit and Bouvier, 2006*), our PCR-based data suggest that both geography and ecology play a role in determining the major biosynthetic components of a microbiome.

The vast majority of AD and KS domain sequences coming from environmental DNA (eDNA) are only distantly related to functionally characterized NRP/PK gene clusters, precluding precise predictions about the specific natural products encoded by the gene clusters from which most amplicons arise. However, in cases where eDNA sequence tags show high sequence similarity to domains found in functionally characterized gene clusters, this information can be used to predict the presence of specific gene cluster families within a specific microbiome. This type of phylogenetic analysis is the basis of the recently developed eSNaPD program, a BLAST-based algorithm for classifying the gene cluster families that are associated with eDNA-derived sequence tags (*Owen et al., 2013*; *Reddy et al., 2014*). When an eDNA sequence tag clades with, but is not identical to, a reference sequence in an eSNaPD-type analysis, it is considered to be indicative of the presence of a gene cluster that encodes a congener (i.e., a derivative) of the metabolite encoded by the reference cluster.

Interestingly, eSNaPD analysis of the data from all sites reveals two distinct types of biomedically relevant natural product gene cluster 'hot spots' within our data (*Figure 2A,B,D*). These include 'specific gene cluster hotspots' and 'gene cluster family hotspots'. Metagenomes from 'specific gene cluster hotspots' are predicted to be enriched for a gene cluster that encodes a congener of the target natural product, while metagenomes from 'gene cluster family hotspots' are predicted to

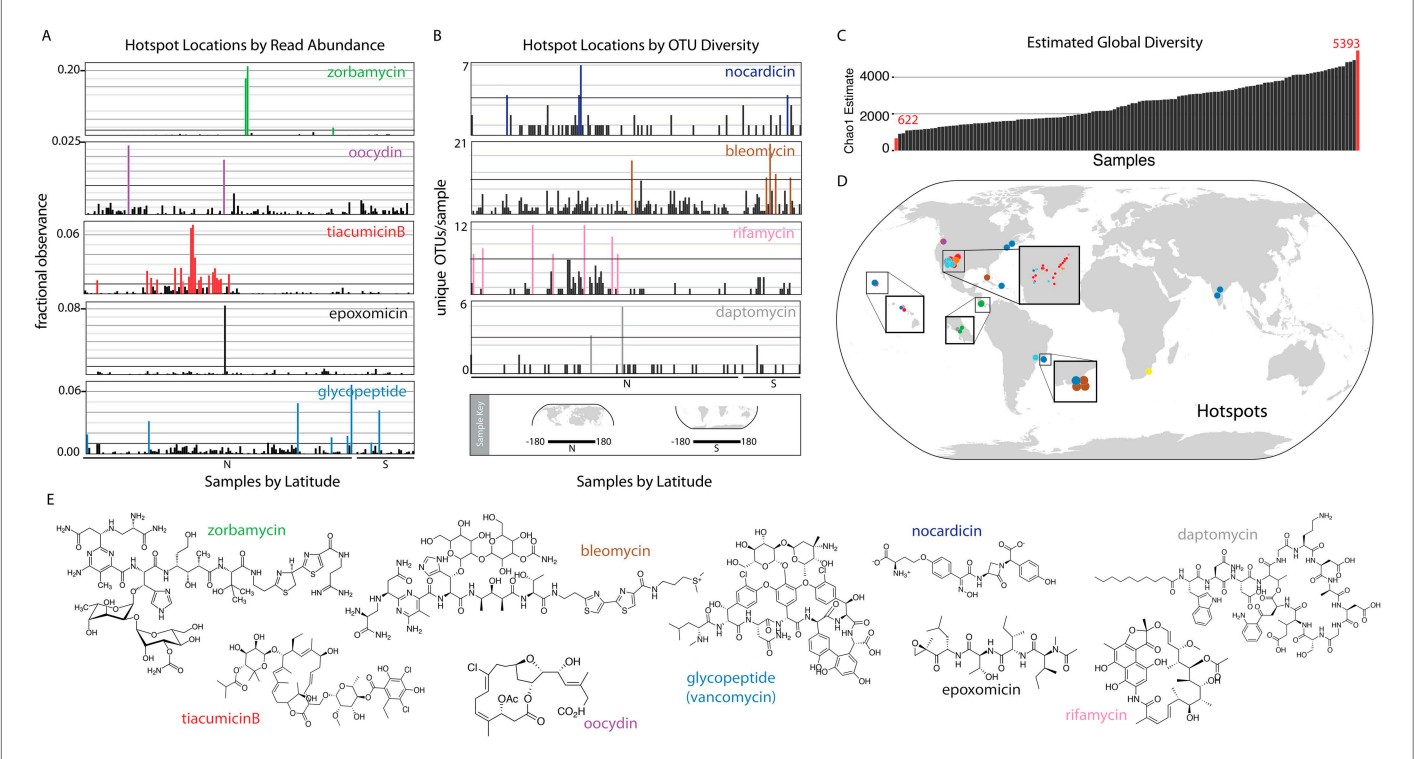

**Figure 2**. Biomedically relevant natural product hotspots and diversity. Hotspot analysis of natural product biosynthetic diversity to identify samples with a high total proportion of reads corresponding to a natural product family of interest (**A** and **D**), the maximum unique OTUs corresponding to a natural product family of interest (**B** and **D**), or the estimated sample biodiversity (**C** and **D**). In **A** and **B** samples are arranged by longitude and hemisphere as is shown in the Sample Key. (**A**) For each sample, sequence reads assigned by eSNaPD are expressed as a percentage of total reads obtained for that sample. A sample is designated a hotspot if more than one percent (0.01; horizontal line) of its reads map to a specific gene cluster. Fractional observance data for five representative gene clusters or gene cluster families (zorbamycin, oocydin, tiacumicinB, epoxomicin, glycopeptides) that show significant sample dependent difference in read frequency are shown. (**B**) Hotspots of elevated gene cluster family diversity can be identified by determining the number of unique OTUs occurring in each sample that, by eSNaPD, map to a natural product gene cluster of interest. Sample specific OTU counts for nocardicin, rifamycin, bleomycin, and daptomycin clusters are shown. Samples containing greater than 50% of the maximum observed OTU value are colored and mapped in (**C**). OTU diversity measurements do not predict the abundance of a specific cluster in a metagenome [as predicted in (**A**)], but instead are used to identify locations where the largest number of congener-encoding clusters may be found. These sites are predicted to be most useful for increasing the structural diversity and therefore potential clinical utility of these medically important families of natural products. (**C**) Estimated diversity of AD/KS reads by sample. AD and KS OTU tables were combined and for each sample the Chao1 diversity metric was calculated at 5000 reads, providing a baseline metric for comparing sample biosynthetic diversity. The average number of unique OTUs observed over 10 rarefactions analyses is shown (also see *Supplementary file 7*). (**D**) Hotspot map of samples identified in **A**, **B** and **C**. (**E**) Representative structures of target molecule families highlighted in **A** and **B**.

encode multiple congeners related to the target natural product. *Figure 2A* shows several of the strongest examples of 'specific gene cluster hotspots' where reads falling into an OTU related to a specific biomedically relevant gene cluster or gene cluster family are disproportionately represented in the sequence data from individual microbiomes. These examples highlight the different enrichment patterns that we observe in the environment—hotspots are either local in nature, consisting of only one or two samples containing sequence reads mapping to the target (epoxomicin, oocydin); regional (tiacumicinB); or global with punctuated increases in diversity (glycopeptides). We would predict 'specific gene cluster hotspots' (*Figure 2D*) are naturally enriched for bacteria that encode congeners of the biomedically relevant target metabolites, thereby potentially simplifying the discovery of new congeners. *Figure 2B* shows examples of 'gene cluster family hotspots', where metagenomes having a disproportionately high number of OTUs mapping to a specific biomedically relevant target molecule family (e.g., nocardicin, rifamycin, bleomycin, and daptomycin families are shown) are highlighted. This analysis identifies specific sample sites, from among those surveyed, that are predicted to contain the most diverse collection of gene clusters associated with a target molecule of interest (*Figure 2B*).

Both types of hotspots should represent productive starting points for future natural product discovery efforts aimed at expanding the structural diversity and potential utility of specific biomedically relevant natural product families.

Biosynthetic domain sequence tag data are not only useful for pinpointing environments that are rich in specific biosynthetic targets of interest but also as a metric for natural product biosynthetic diversity in general. As only a small fraction (5–10%) of total AD and KS sequences can be confidently assigned by the eSNaPD algorithm, samples showing the largest collection of unique OTUs (at a common sequencing depth) might be expected to contain the most diverse collection of novel biosynthetic gene clusters (*Figure 2C*) and therefore be the most productive sites to target for future novel molecule discovery efforts. Once normalized for sequencing depth, the number of unique KS and AD sequence tags observed per collection site differs by almost an order of magnitude between environments (*Figure 2C*), with the most diverse samples mapping to Atlantic forest and Desert environments (*Figure 2C,D* teal spots, *Supplementary file 7*).

The development of cost effective high-throughput DNA sequencing methodologies and powerful biosynthesis focused bioinformatics algorithms allow for the direct interrogation and systematic mapping of global microbial biosynthetic diversity. Our analyses of 100s of distinct soil microbiomes suggests that geographic distance and local environment play important roles in the sample-to-sample differences we detected in biosynthetic gene populations. As variations in biosynthetic gene content are expected to correlate with variations in the small-molecule producing capabilities of a microbiome, the broader implication of these observations from a drug discovery perspective is that the dominant biosynthetic systems of geographically distinct soil microbiomes are expected to encode orthogonal, largely unexplored collections of natural products. Taken together, our biosynthetic domain hotspot and OTU diversity analyses represent a starting point in the creation of a global natural products atlas that will use sequence data to guide natural product discovery in the future. Based on the historical success of natural products as therapeutics, microbial 'biosynthetic dark matter' is likely to hold enormous biomedical potential. The key will be learning how to harvest molecules encoded by the biosynthetic diversity we are now able to find through sequencing.

## Materials and methods

### Soil collection

Soil from the top 6 inches of earth was collected at unique locations in the continental United States, China, Brazil, Alaska, Hawaii, Costa Rica (*Brady and Clardy, 2004*), Ecuador, the Dominican Republic, Australia, Tanzania and South Africa. The full sample table is available in *Supplementary file1*.

### Soil DNA extraction

To reduce the potential for cross contamination, DNA was extracted from soil using a simplified version of our previously published DNA isolation protocol (*Brady, 2007*; *Reddy et al., 2012*). The modified protocol was as follows: 250 grams of each soil sample was incubated at 70°C in 150 ml of lysis buffer (2% sodium dodecyl sulfate [wt/vol], 100 mM Tris–HCl, 100 mM EDTA, 1.5 M NaCl, 1% cetyl trimethyl-ammonium bromide [wt/vol]) for 2 hr. Large particulates were then removed by centrifugation (4000×*g*, 30 min), and crude eDNA was precipitated from the resulting supernatant with the addition of 0.6 vol of isopropyl alcohol. Precipitated DNA was collected by centrifugation (4000×*g*, 30 min), washed with 70% ethanol and resuspended in a minimum volume of TE (10 mM Tris, 1 mM EDTA [pH 8]). Crude environmental DNA was passed through two rounds of column purification using the PowerClean system (MO BIO, Carlsbad, California). Purified environmental DNA was then diluted to 30 ng/µl and archived for use in PCR reactions.

### PCR amplification

Degenerate primers targeting conserved regions of AD [A3F (5'-GCSTACSYSATSTACACSTCSGG) and A7R (5'-SASGTCVCCSGTSCGGTA) (*Ayuso-Sacido and Genilloud, 2005*)] and KS [degKS2F.i (5'-GCIATGGAYCCICARCARMGIVT) and degKS2R.i (5'-GTICCIGTICCRTGISCYTCIAC) (*Schirmer et al., 2005*)] domains were used to amplify gene fragments from crude eDNA. Forward primers were designed to contain a 454 sequencing primer (CGTATCGCCTCCCTCGCGCCATCAG) followed by a unique 8 bp barcode that allowed simultaneous sequencing of up to 96 different AD- or KS- samples in a single GS-FLX Titanium region. PCR reaction consisted of 25 µl of FailSafe PCR Buffer G (Epicentre, Madison, Wisconsin), 1 µl recombinant *Taq* Polymerase (Bulldog Bio, Portsmouth, New Hampshire),

1.25 µl of each primer (100 mM), 14.5 µl of water and 6.5 µl of purified eDNA. PCR conditions for AD domain primers were as follows: 95°C for 4 min followed by 40 cycles of 94°C for 0.5 min, 67.5°C for 0.5 min, 72°C for 1 min and finally 72°C for 5 min. PCR conditions for KS domain primers were as follows: 95°C for 4 min followed by 40 cycles of 54°C for 40 s, 56.3°C for 40 s, 72°C for 75 s and finally 72°C for 5 min. PCR reactions were examined by 2% agarose gel electrophoresis to determine the concentration and purity of each amplicon. Amplicons were pooled in equal molar ratios, gel purified using the Invitrogen eGel system and DNA of the appropriate size was recovered using Agencourt Ampure XP beads (Beckman Coulter, Brea, California). Amplicons were sequenced using the 454 GS-FLX Titanium platform. Raw flowgram files from 454's shotgun processing routine were used for downstream analysis.

## Processing 454 data

Raw reads were assigned to samples using the unique primer barcodes and filtered by quality (50 bp rolling window PHRED cutoff of 20) using Qiime (version 1.6) (*Caporaso et al., 2010*). USEARCH (version 7), which implements the improved UPARSE clustering algorithm (*Edgar, 2013*), was used to remove Chimeric sequences with the default 1.9 value of the de novo chimera detection tool. UPARSE clustering requires all sequences to be of the same length. In an effort to balance read quality and abundance with the ability to phylogenetically discriminate gene clusters we used 419 bp as our read length cutoff. The trimmed fasta file was then clustered to 5% to compensate for sequencing error and natural polymorphism that is often observed in gene clusters found in natural bacterial populations. Clustering proceeded as per the USEARCH manual by clustering at a distance of 3% and using representative sequences from each cluster to cluster again at 5%. The resulting '5%' AD and KS OTU tables were used for all subsequent rarefaction and diversity analyses. Read and OTU counts available in *Supplementary file 2*.

## Rarefaction and diversity analyses

To assess global AD and KS diversity in our sample set we sought to assess the global number of AD and KS domains we might expect to see if all of our data had been generated from a single sample. To do this, all reads assigned to an OTU were consolidated to generate a single-column OTU table where each row contains the sum of all sequences assigned to that OTU from any of the 185 samples. To assess the global diversity we subsampled this table at multiple depths using Qiime (*Caporaso et al., 2010*) and used the Chao1 formula to estimate the expected number of OTUs at this depth. This rarefaction analysis was performed ten times at each subsampling depth (*Figure 1A,C*; *Supplementary files 3, 4*) and the curves were fit to the data using the following equation: $y = 1 + \log(x) + \log(x^2) + \log(x^3)$ where x is the read value and y is the Chao1 diversity.

Ecological distances are calculated using the Jaccard [$1 - (OTU_{A\&B})/(OTU_A + OTU_B - OTU_{A\&B})$] or inverse Jaccard metric (*Oksanen et al., 2013*) and geographic distances were calculated using great circle (spherical) distance derived from the latitude/longitude values of each set of points (*Bivand and Pebesma, 2005*) (*Supplementary file 5*). Pairwise ecological and geographic distances were used to create *Figure 1B,D*. Network plots of subsamples (*Figure 1G,H*) were generated using Phyloseq (*McMurdie and Holmes, 2013*) to calculate the intersample Jaccard distance. As expected, the strongest relationships are observed between sample proximity controls where soils were collected approximately 10 meters from one another and processed independently, demonstrating that closely related samples do in fact group together in our analysis pipeline.

## Assignment of AD and KS domains to known gene clusters

AD and KS amplicon reads were assigned to known biosynthetic gene clusters using the eSNaPD algorithm at an e-value cutoff of $10^{-45}$ (*Reddy et al., 2014*). At this threshold eSNaPD has been used to successfully assign-and-recover gene clusters that encode congeners of multiple natural product families using only the sequence from a single domain amplicon (*Owen et al., 2013*; *Chang and Brady, 2014*; *Kang and Brady, 2014*). NRPS/PKS clusters typically have multiple KS or AD domains. Hits to all domains in a cluster were aggregated in our analyses. Data for eSNaPD hits broken down by sample and molecule are included as *Supplementary file 6*.

## Hotspot analysis

AD and KS OTU tables were analyzed for the presence of eSNaPD hits. For each sample the abundance of each eSNaPD hit (i.e., a particular molecule) was calculated as either a percentage of total reads (*Figure 2A,C*) or as the total number of unique OTUs assigned to the molecule that were found in that sample (*Figure 2B,C*), or as the total number of OTUs mapped to a molecule in each sample.

In the read-based hotspot analysis, the number of reads assigned by eSNaPD to a specific gene cluster is expressed as a fraction of total per sample reads: (reads-to-cluster-of-interest)/total sample reads). In the OTU-based hotspot analysis we calculated the number of unique eSNaPD assigned OTUs found in each sample that map to a specific gene cluster. The full eSNaPD dataset is available in *Supplementary file 6*. To compare global biosynthetic diversity of each sample, the AD and KS OTU tables were combined and for each sample they were subsampled ten times to a depth of 5000 reads. The Chao1 diversity metric was calculated for each sample and the average was used to compare the expected biodiversity in different samples at the same sampling depth (*Figure 1C*, *Supplementary file 7*).

## Acknowledgements

This work was supported by National Institutes of Health grant number GM077516 (SFB), and F32 AI110029 (ZCP). SFB is a Howard Hughes Medical Institute Early Career Scientist. Brazilian research was supported by São Paulo Research Foundation (FAPESP) grant #2011/50869-8. MTP is a research fellow of the Conselho Nacional de Desenvolvimento Cientifico e Tecnologico (CNPq). DOG was supported by the Rio de Janeiro Research Foundation (FAPERJ) grant #E-26/110.281/2012 and CNPq grant #477509/2013-4. The authors would also like to acknowledge the following people and institutions for assistance in collecting samples: Rafael Bonante, Samyr Soares Viana, Vitor de Carli, Ronaldo de Carli, Erin Bishop, Vanessa Kowalski, The Instituto National de Bioversidad, Serengeti Genesis, and the University of Dar Es Salaam.

## Additional information

### Funding

| Funder | Grant reference number | Author |
|---|---|---|
| National Institutes of Health (NIH) | F32 AI110029 | Zachary Charlop-Powers |
| Rio de Janeiro Research Foundation | #E-26/ 110.281/2012 | Denise O Guimarães |
| Conselho Nacional de Desenvolvimento Científico e Tecnológico | #477509/2013-4 | Denise O Guimarães |
| Howard Hughes Medical Institute (HHMI) | Early Career Scientist | Sean F Brady |
| National Institutes of Health (NIH) | GM077516 | Sean F Brady |
| São Paulo Research Foundation (FAPESP) | #2011/50869-8 | Denise O Guimarães, Monica T Pupo |
| Conselho Nacional de Desenvolvimento Científico e Tecnológico | Research Fellow | Monica T Pupo |

The funders had no role in study design, data collection and interpretation, or the decision to submit the work for publication.

### Author contributions

ZC-P, Conception and design, Acquisition of data, Analysis and interpretation of data, Drafting or revising the article; JGO, MAT, Acquisition of data, Analysis and interpretation of data, Drafting or revising the article; BVBR, Analysis and interpretation of data, Drafting or revising the article; DOG, Acquisition of data, Analysis and interpretation of data, Contributed unpublished essential data or reagents; UAF, PS, ZF, Acquisition of data, Drafting or revising the article, Contributed unpublished essential data or reagents; MTP, TBC, Acquisition of data, Drafting or revising the article, Contributed unpublished essential data or reagents; SFB, Conception and design, Analysis and interpretation of data, Drafting or revising the article

### Author ORCIDs

Zachary Charlop-Powers, http://orcid.org/0000-0001-8816-4680
Ulysses A de Frias, http://orcid.org/0000-0002-3739-2779
Monica T Pupo, http://orcid.org/0000-0003-2705-0123

## Additional files

### Supplementary files

• Supplementary file 1. Sample Location Data.

• Supplementary file 2. Sample Read and 95% OTU Count.

• Supplementary file 3. Adenylation Domain Rarefaction Data (*Figure 1A*).

• Supplementary file 4. Ketosynthase Domain Rarefaction Data (*Figure 1C*).

• Supplementary file 5. Pairwise Sample Distances. Great Circle Distance and Jaccard Distance for AD and KS Amplicons.

• Supplementary file 6. eSNaPD Hits Broken Down by Sample and Molecule.

• Supplementary file 7. Per Sample Chao1 Biodiversity Estimates at a Rarefaction Depth of 5000 Reads.

### Major dataset

The following dataset was generated:

| Author(s) | Year | Dataset title | Dataset ID and/or URL | Database, license, and accessibility information |
|---|---|---|---|---|
| Brady Sean, et al., | 2014 | Secondary Metabolite Metagenome Amplicons | http://www.ncbi.nlm.nih.gov/bioproject/PRJNA258222 | Public Domain. |

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
