## [Decision Letter]

Thank you for sending your work entitled “Global Biogeography of Bacterial
Secondary Metabolism” for consideration at *eLife*. Your article
has been favorably evaluated by Ian Baldwin (Senior editor) and 4 reviewers, one of
whom, Jon Clardy, is a member of our Board of Reviewing Editors.

The Reviewing editor and the other reviewers discussed their comments before we reached
this decision, and the Reviewing editor has assembled the following comments to help you
prepare a revised submission.

The reviewers felt that the paper raised a significant point regarding the widespread
understanding that “everything is everywhere, the environment selects”.
However, in its present form, the tone of the article overstated the extent to which the
current study discredited this understanding and suggested a number of modifications
that would tone down the conclusions. These began with the title, and ”Global
Biogeographical Sampling of Bacterial Secondary Metabolism“ was suggested as more
consistent with the actual research presented. Most of the comments focused on: the
small sample size, and the degree to which it broadly surveyed potential ecological
sites; the restricted focus on NRPS and KS pathways, which in turn select for a subset
of bacteria (with an explicit reference to the recent Fischbach paper in Cell); and the
ability of sequence tag data (as opposed to deeper sequencing) to serve as a metric for
natural product biosynthetic diversity in general.

The reviewers agree that even with these limitations the data presented offer a
significant challenge to the reigning “everything is everywhere” paradigm,
but they don't completely overthrow it.

A concern about the degree with which the “crowd sourcing” approach
aligned with the Nagoya Protocol on Biological Diversity was also raised.

---

## [Author Response]

*1a) The reviewers felt that the paper raised a significant point regarding the
widespread understanding that “everything is everywhere, the environment
selects”. However, in its present form, the tone of the article overstated the
extent to which the current study discredited this understanding and suggested a
number of modifications that would tone down the conclusions*.

We agree that our data are insufficient to overturn the “everything is everywhere
but the environment selects” hypothesis, and it was not our intention to
specifically address that topic in this manuscript. On one hand, we see that gene
populations are more similar from similar environments suggesting a selective effect. On
the other hand, geographically distant sites from similar environments have relatively
few OTUs in common. While the “everything/everywhere” theory is of immense
interest in understanding the ecology and natural history of microbial communities, we
have tried to limit ourselves to empirical descriptions of the populations that suggest
meaningful differences among sample locations irrespective of how those differences came
to be. From the perspective of functional metagenomics, the origin of these differences
is less important than being able to describe and utilize them.

To address the relationship between our data and the Baas-Becking hypothesis directly,
in the manuscript we have added the following sentence to the third paragraph of the
Results and discussion section:

“Although it is possible that at a much greater sampling depth all AD and KS
domains will be found at all sites as predicted by Baas-Becking’s
‘everything is everywhere but the environment selects’ hypothesis of
global microbial distribution, our data suggests a strong dependence on geography and
ecology in determining the major biosynthetic components of a microbiome.”

We have also made changes to the wording throughout the manuscript, which are intended
to tone down the conclusions of the manuscript and make it clear that our statements are
a reflection of the samples collected and are not necessarily trends across the
globe.

*1b) These began with the title, and ”Global Biogeographical Sampling of
Bacterial Secondary Metabolism“ was suggested as more consistent with the
actual research presented*.

The title has been changed to “Global Biogeographic Sampling of Bacterial
Secondary Metabolism” as suggested.

*1c) Most of the comments focused on: the small sample size, and the degree to
which it broadly surveyed potential ecological sites*.

We entirely agree that a strong statement on the global abundance and distribution will
require sampling the Earth’s surface at a much greater density than we were able
to achieve in this study. Nonetheless, this is the largest survey to date targeting
NRPS/PKS biosynthetic genes and we believe that there are a number of trends that emerge
at this degree of sampling, most notably the large differences between microbiomes
around the world with the intriguing corollary that there are large numbers of gene
clusters we may not have investigated for their biosynthetic potential. As outlined
above we have now clarified in a number of places in the manuscript that all conclusions
drawn reflect the dataset we analyzed and not the Globe as a whole.

*1d) The restricted focus on NRPS and KS pathways, which in turn select for a
subset of bacteria (with an explicit reference to the recent Fischbach paper in
Cell)*.

The NRPS and PKS biosynthetic families are a natural starting point for
metagenome-driven drug discovery for two reasons: first, they are overrepresented as
pharmacologically active agents, and, second, their conserved biosynthetic domains allow
a PCR-based approach to detect amplicons belonging to clusters that encode a diverse set
of compounds. The comprehensive overview of sequenced genomes that was recently
published in Cell, “Insights into Secondary Metabolism from a Global Analysis of
Prokaryotic Biosynthetic Gene Clusters,” shows that there are a handful of
biosynthetic systems that are repeatedly found in the genomes of known bacteria that the
authors characterize as follows: Saccharide, Other, NRPS, PKS/Fatty Acid, Hybrid
(NRPS/PKS), Ribosomal Peptides, and Terpenes. These cluster families are the usual
suspects of secondary metabolism. Although the paper highlights the existence of
bacterial gene cluster families whose natural products are not known, the dominant
players seen in this data analysis remain these well characterized gene cluster
families. These same trends are seen in an extensive gene cluster analysis recently
published by Bill Metcalf’s group in Nature Chemical Biology entitled “A
roadmap for natural product discovery based on large-scale genomics and
metabolomics”. The large set of saccharide-encoding gene clusters reported in one
survey paper likely significantly overestimates the role of sugar metabolism in natural
product biosynthesis by including saccharides with functions with a structural rather
than signaling/defensive role (cellulose, O-antigens, Capsular polysaccharides, etc.).
Therefore, we believe all recently reported gene cluster surveys essentially confirm the
principle upon which our study is based, namely, NP diversity is dominated by a small
group of conserved families and that examination of these families is likely to be the
most informative avenue for studying chemical diversity in the environment. We have
chosen NRPS/PKS as they are largest of these dominant biosynthetic systems although it
will be a natural extension of this work for our lab and others to extend it to other
families as well.

*1e) The ability of sequence tag data (as opposed to deeper sequencing) to serve
as a metric for natural product biosynthetic diversity in general*.

We agree with the reviewers that long contigs with greater contextual information about
the bacterial gene clusters would allow for a better analytical power but disagree that
deeper sequencing would provide that information as assembling metagenomes remains a
significant technical bottleneck in metagenome research. Currently, there is no easy way
to assemble a heterogeneous population of soil-derived genomes using shotgun methods
despite the many groups spending a lot of time and money trying to develop algorithms or
technical solutions to the problem. Contemporary state-of-the-art metagenome assemblies
from complex microbiomes result in incomplete assembly and short contigs. The scale of
sequencing required for generating even very low quality information from complex
microbiomes makes it impractical to employ this technique for screening large numbers of
unique microbiomes. Because of the shortcomings of shotgun-methods in soil metagenomes,
we have championed the use of sequence tags to bypass these technical problems. Unlike
whole genome sequencing methodologies, this approach does not rely on the analysis of
complete biosynthetic clusters. Instead, it uses individual next-generation sequencing
reads from PCR amplicons of conserved biosynthetic domain gene sequences (termed Natural
Product Sequence Tags, NPSTs) to predict gene content and chemical output of NP gene
clusters, in a fashion analogous to reconstructing the phylogeny of entire organisms
using 16S rRNA sequences. Although shotgun-sequencing approaches have been useful for
guiding the identification of clusters in individual genomes and small, endosymbiont
metagenomes, their application to more complex metagenomes has been very limited. In
fact, in a direct comparison of biosynthetic domain detection from metagenomic samples,
this PCR-based method was shown to be 10 to 100 times more sensitive than shotgun
sequencing in identifying unique sequences.

*2) The reviewers agree that even with these limitations the data presented offer
a significant challenge to the reigning ”everything is everywhere“
paradigm, but they don't completely overthrow it*.

Please see comment 1a.

*3) A concern about the degree with which the ”crowd sourcing“
approach aligned with the Nagoya Protocol on Biological Diversity was also
raised*.

Although crowd sourcing was used to obtain samples for this study, these samples were
almost exclusively from within the United States. Of the foreign (non-US) samples
collected for this study, the vast majority of these samples were collected and
processed by scientists from the country of origin. Additional non-United States samples
represent eDNA samples originally collected in earlier international collaborative
studies. In addition to our collection methods being of a collaborative nature with
international scientists, our sample handling methods are designed to ensure that
neither live bacteria nor cloneable eDNA is ever collected or available at the end of
the experiment. It is therefore not possible to recover commercializable material (e.g.,
bacterial gene clusters, live bacteria or small molecules) from any foreign sample
processed in this study. As no archiveable material was collected for this work, the
gene fragments sequenced in this study are only useful for conducting comparative
ecological studies like those outlined here, and they do not permit the recovery of any
functional genetic material that might be of commercial value.